# Adsorption of Atrazine from Synthetic Contaminated Water Using a Packed-Bed Column with a Low-Cost Adsorbent (*Moringa oleifera* Lam.)

Rosângela Bergamasco [1,*], Daniel Mantovani [1], Alexandre Diório [1], Charleston de Oliveira Bezerra [1], Heloise Beatriz Quesada [1], Gessica Wernke [1], Márcia Regina Fagundes-Klen [2] and Luís Fernando Cusioli [1]

[1] Department of Chemical Engineering, State University of Maringá, Maringá 87020-900, Paraná, Brazil; daniel26mantovani@gmail.com (D.M.); adiorio2@uem.br (A.D.); charleston10@gmail.com (C.d.O.B.); gessica.wernke@hotmail.com (G.W.); luiscusioli@gmail.com (L.F.C.)

[2] Department of Chemical Engineering, State University of West Parana, Toledo 85903-000, Parana, Brazil; fagundes.klen@gmail.com

* Correspondence: ro.bergamasco@hotmail.com; Tel.: +55-44-3011-4782

**Abstract:** Atrazine (ATZ) is an herbicide used for increased food production due to its weed and pesticide control capacity in different crops. However, ATZ is a chemical compound that is harmful to the environment and human health, and, unfortunately, it has been detected in surface and groundwater. Therefore, the aim of this paper was to perform the adsorption of Atrazine from a synthetically contaminated water sample using a packed-bed column with a low-cost adsorbent prepared from *Moringa oleifera* Lam. seeds. The synthesized adsorbent presented an increase in the surface specific area ($S_{BET}$) of 37% in comparison with the *in natura* material. The effects of the peristaltic pump flow rate (Q), concentration of the ATZ inlet ([ATZ]$_{inlet}$) solution, and bed height (H) were studied, with the highest percentage of ATZ removed through the adsorption column (50, 0%) obtained with a packed-bed column with H = 13 cm bed height, Q = 1 mL/min, [ATZ]$_{inlet}$ = 2.0 mg/L, pH = 5.0, a breakthrough time of 25 min, and a saturation time of 420 min. The logistic model was used to best fit the experimental data with an $R^2$ > 0.99, and the Bohart–Adams, Thomas, and Yoon–Nelson models were used to explain and analyze the obtained effects in the continuous adsorption of ATZ. Therefore, the *Moringa oleifera* Lam. seeds provided a low-cost adsorbent for the continuous adsorption of the herbicide Atrazine in a packed-bed column.

**Keywords:** herbicide; low-cost adsorbent; CEC removal; water remediation

## 1. Introduction

Developed in the late 1950s, Atrazine (ATZ) became one of the most used herbicides worldwide for food production and is applied over agricultural and non-agricultural lands [1]. The global market for ATZ is expected to reach USD 2.58 billion by 2024, an increase of 6.8% from 2019. ATZ is a chemical carcinogenic to humans [2] and is classified as Group 2B with the IUPAC name of 6-chloro-N2-ethyl-N4-isopropyl-1,3,5-triazine-2,4-diamine [3].

The intensive use of ATZ in agricultural activities aims to increase crops and harvest productivity due to the herbicide's efficiency in weed and pest control, cheapness, and high effectiveness, being the second most used herbicide worldwide [4]. Therefore, ATZ has been applied to different crops such as corn, soy, sugarcane, and others [5,6]. If ATZ use can increase food production, its persistence in soils and toxicity may cause diseases in humans [7,8] and pose serious environmental risks [1,9]. However, due to the rapid population growth, the use of ATZ and other herbicides is a must to address the food shortage [10].

There are three pathways for the ATZ to enter the human body and harm its health: (i) inhalation of suspended ATZ particles and/or (ii) dermal absorption during pulverization (spraying) or transport, and (iii) direct ingestion of contaminated soil and/or water [7]. Water contamination by ATZ is an important problem to be addressed due to its human health risks and toxicity to other organisms [11]. It extensively explained the toxic effects of ATZ on humans, plants, animals, and microorganisms. Those effects include, but are not limited to, endocrine disruptors [12], carcinogenic effects at low concentrations [13], and an increase in honey bee mortality.

Approximately, 0.2–2 kg/hectare of herbicides are used in farmlands in developing or developed countries [14]. The eroded-contaminated soil and the free chemicals are carried out by the rainfall or irrigation waters, resulting in surface and groundwater contamination [5,6]. In this sense, in Brazil, the National Water Agency [15] stated that water body contamination due to herbicides/pesticides is the second-largest cause, behind only domestic sewage contamination. Therefore, ATZ was classified by the Brazilian Government as a chemical that poses health risks [16] but is still used in Brazil and other countries [17]. It is important to note that the United States Environmental Protection Agency limits the ATZ concentration in water to a threshold of $3 \ \mu g \ L^{-1}$, but residual ATZ concentrations in aquatic environments were detected varying between $0.5 \ \mu g \ L^{-1}$ and $22 \ mg \ L^{-1}$ [18]. In addition, the European Union banned ATZ use in 2004; and Germany, which prohibited ATZ use in 1991, still found traces of it in water samples twenty years later [1].

In any case, despite its low water solubility, water remediation for ATZ removal is necessary, and different technologies have been applied for ATZ removal [10,19], such as advanced oxidation processes [20], membrane nanofiltration [21], and biodegradation processes, but adsorption with different adsorbents has stood out [18]. Adsorption is a promising technology for ATZ removal from contaminated water due to its universal applicability, simplicity of operation and design, economic efficiency, and the fact that no harmful or toxic by-products are formed [3].

Previously, a low-cost adsorbent prepared from *Moringa oleifera* Lam. seed husks was applied in a discontinuous adsorption process for ATZ removal [22]. The novelty of this research is to intensify the adsorption process by employing a continuous one, aiming to determine the process conditions for the highest ATZ removal. Therefore, the aim of this paper was to perform the adsorption of Atrazine from synthetically contaminated water using a packed-bed column with a low-cost adsorbent prepared from *Moringa oleifera* Lam. seeds.

## 2. Materials and Methods

### 2.1. Materials

Atrazine (ATZ, 6-chloro-N2-ethyl-N4-isopropyl-1,3,5-triazine-2,4-diamine) was provided by Nortox® enterprise (Arapongas, Brazil). The synthetically contaminated water was prepared by dissolving different masses of ATZ in deionized water, and the ATZ concentration was determined by means of a spectrophotometer (UV/Vis, HACH DR 500) at $\lambda = 222$ nm. The seeds from *Moringa oleifera* Lam. were provided by the Federal University of Sergipe (Aracajú, Brazil).

### 2.2. Adsorbent Treatment

Initially, the *Moringa oleifera* Lam. seeds were manually peeled off, washed with deionized water at 45 °C, and immersed in a 0.1 M methanol solution for 4 h at an m/V ratio of 1:5. Then, the pre-treated seeds were transferred to a 0.1 M nitric acid solution for 1 h. The methanol and acid treatment aimed to remove undesired inorganic ions and impurities linked to the adsorption sites [23] and was dried in a micro-processed air oven (Digital Timer SX CR/42) for 12 h. Finally, the material was placed in an oven (Jung 10.012) at 300 °C for 1 h [24]. After cooling, the material was ground and sieved between 0.35 and 0.50 mm, thus being named adsorbent.

### 2.3. Adsorbent Characterizations

The superficial morphology of the adsorbent and *Moringa oleifera* Lam. seeds was characterized by scanning electron microscopy (SEM; FEI Quanta 200). The surface modifications due to the chemical and thermal treatment were verified by means of Fourier transform-infrared (FTIR) analysis using a spectrophotometer (Vertex 70v Bruker) in the wavelength range of 400 to 4000 cm$^{-1}$. The amorphous structure of the material was analyzed by X-ray diffraction (XRD; Shimadzu LabX 6000) with CuK $\alpha$-radiation ($\lambda$ = 1.54056), 40 kV and 30 mA, rotational rate of 2° 2$\theta$ min$^{-1}$ from $5 \leq 2° \theta \leq 80$, and acquisition time of 1 s. The specific surface area and pore volumes were obtained by means of N$_2$-physisorption at 77 K (Micrometrics ASAP 2020) with the Brunauer–Emmet–Teller method [25] for the specific surface area determination and the single-point method for the total pores volume at p $\times$ p$_0^{-1}$ = 1. The "11-points methodology" from Regalbuto and Robles [26] was used for the pH of point of zero charge (pH$_{PZC}$) determination. The zeta potential was measured in a particle analyzer (Delsa TMNanoC, Beckman Coulter). Solutions of HCl (0.1 M) and NaOH (0.1M) were used for pH adjustment when needed.

### 2.4. Packed-Bed Column Adsorption Experiments

The adsorption of ATZ was performed by packing a 26 cm-high glass column with different amounts of the prepared adsorbent. The column inner diameter was 0.9 cm, and the column packing followed the "slurry method" [27]. Briefly, $\frac{1}{4}$ of the column was filled with glass beads fixed by a polyamide fabric, then different amounts of adsorbent were packed, followed by glass beads and another polyamide fabric as described in Figure 1. The column inlet was fed with the atrazine-contaminated solution upward through the column by a peristaltic pump at different flow rates. After running through the column, samples were manually collected at different times, and the residual concentration of Atrazine was determined.

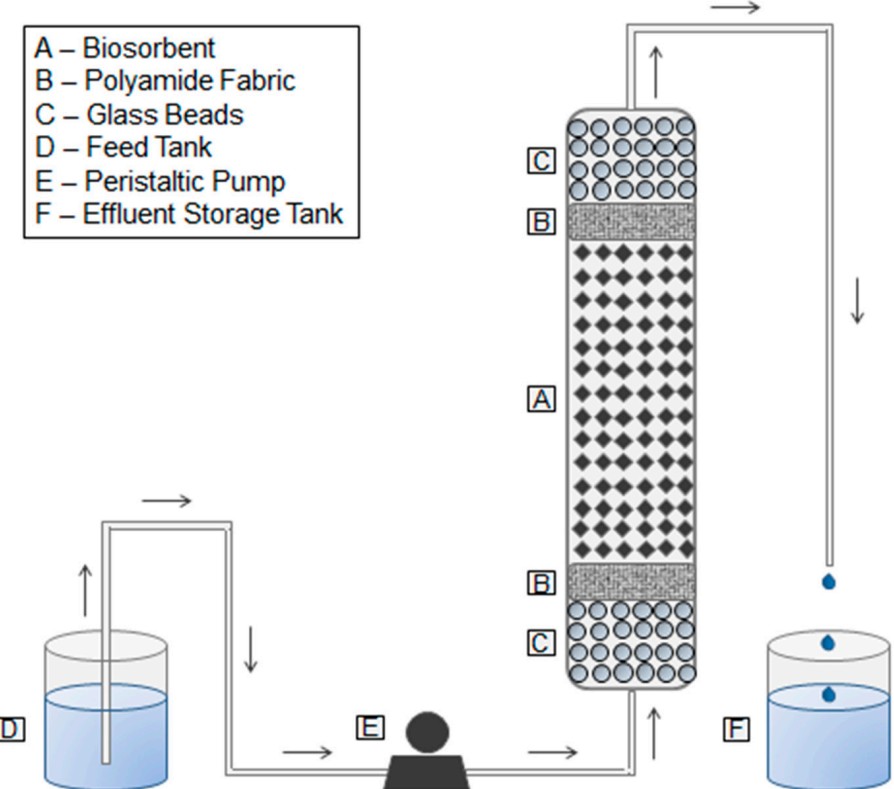

A – Biosorbent
B – Polyamide Fabric
C – Glass Beads
D – Feed Tank
E – Peristaltic Pump
F – Effluent Storage Tank

**Figure 1.** Schematic diagram of the lab-scale packed-bed column for adsorption.

### 2.4.1. Effect of the Flow Rate

The peristaltic pump flow rate (Q) varied (1, 2, and 3 mL min$^{-1}$) in order to analyze the effect of the flow rate on the adsorption process using the adsorbent prepared with the seeds of *Moringa oleifera* Lam. The ATZ inlet concentration (2 mg L$^{-1}$, pH = 5.0) and the bed height (13 cm) were fixed during the experiments.

### 2.4.2. Effect of the Bed Heigh

The effect of the bed height (H) was analyzed by packing the column with 5, 9, and 13 cm of adsorbent. The ATZ inlet concentration was fixed (2 mg L$^{-1}$) with a solution pH of 5.0 and a fixed flow rate of 1 mL min$^{-1}$.

### 2.4.3. Effect of the Inlet Concentration

The effect of the ATZ inlet concentration on the adsorption was verified for 2, 5, and 8 mg L$^{-1}$. The flow rate was fixed at 1 mL min$^{-1}$, pH 5.0, and bed height of 13 cm.

### 2.5. Fixed-Bed Adsorption

The column adsorption experiments were performed at ambient temperature (25 °C) in duplicates. The breakthrough time ($t_b$) was determined as the time necessary for the residual concentration of ATZ ([ATZ]$_{outlet}$) to be equal to 10% of the initial ATZ concentration ([ATZ]$_{inlet}$), i.e., $t_b$ occurs when [ATZ]$_{outlet}$/[ATZ]$_{inlet}$ = 0.10. Similarly, the saturation time ($t_s$) was defined as the adsorption time for [ATZ]$_{outlet}$/[ATZ]$_{inlet}$ = 0.90 [28].

The performance of the column adsorption was analyzed in terms of the adsorption capacity of the column at breakthrough time ($q_b$), determined by Equation (1), and the maximum column capacity ($q_t$) at total saturation time, according to Equation (2).

$$q_b = \frac{QA}{1000} = \frac{Q}{1000} \int_{t=0}^{t=t_b} [ATZ]_b dt \tag{1}$$

where $q_b$ is the mass (mg) of ATZ adsorbed at breakthrough time ($t_b$, min); Q is the volumetric flow rate (mL min$^{-1}$); A is the area above the breakthrough curve; and [ATZ]$_b$ = [ATZ]$_{inlet}$−[ATZ]$_{outlet}$ (mg L$^{-1}$) until [ATZ]$_{outlet}$/[ATZ]$_{inlet}$ = 0.10.

$$q_t = \frac{QA}{1000} = \frac{Q}{1000} \int_{t=0}^{t=t_{total}} [ATZ]_s dt \tag{2}$$

where $q_t$ is the total mass (mg) of ATZ adsorbed at total saturation time ($t_{total}$, min); Q is the volumetric flow rate (mL min$^{-1}$); A is the area above the total saturation curve; and [ATZ]$_s$ = [ATZ]$_{inlet}$−[ATZ]$_{outlet}$ (mg L$^{-1}$) until [ATZ]$_{outlet}$/[ATZ]$_{inlet}$ = 0.90.

The maximum adsorption capacity of the adsorbent ($q_e$, mg g$^{-1}$), defined as the mass of ATZ adsorbed per unit mass of adsorbent, was determined by Equation (3).

$$q_e = \frac{q_t}{M} \tag{3}$$

where M is the total mass of adsorbent (g) packed in the column.

The total mass of ATZ transported through the adsorption column ($W_t$, mg) was calculated according to Equation (4).

$$W_t = \frac{[AZT]_{inlet} \cdot Q \cdot t_s}{1000} \tag{4}$$

Finally, the percentage of ATZ removed through the adsorption column ($\eta$, %) was calculated by Equation (5).

$$\eta = \frac{q_t}{W_t} 100 \tag{5}$$

*2.6. Column Adsorption Modelling*

The logistic model, expressed in Equation (6), was fitted to the experimental data according to [29].

$$\text{Ln} \left( \frac{[\text{ATZ}]_{\text{inlet}}}{[\text{ATZ}]_{\text{outlet}}} - 1 \right) = a - bt \tag{6}$$

where a and b are the logistic model coefficients (intercept and slope, respectively).

Depending on their values, the coefficients a and b are able to represent three column adsorption models: (i) Bohart–Adams model; (ii) Thomas model; and (iii) Yoon–Nelson model. These models and their parameters are displayed in Equations (7)–(9), respectively.

$$a = \frac{k_{\text{BA}}N_0H}{v}, b = k_{\text{BA}}[\text{ATZ}]_{\text{inlet}} \tag{7}$$

$$a = \frac{k_T q_e M}{Q}, b = k_T[\text{ATZ}]_{\text{inlet}} \tag{8}$$

$$a = k_{\text{YN}}\tau, b = k_T[\text{ATZ}]_{\text{inlet}} \tag{9}$$

where $k_{\text{BA}}$ is the Bohart–Adams kinetic constant (L mg$^{-1}$ min$^{-1}$); $N_0$ is the saturation concentration (mg L$^{-1}$); H is the bed height (cm); v is the linear velocity (cm min$^{-1}$) determined by dividing the volumetric flow (cm$^3$ min$^{-1}$) by the column cross-section area (cm$^2$); $k_T$ is the Thomas model rate constant (L mg$^{-1}$ min$^{-1}$); $k_{\text{YN}}$ is the Yoon–Nelson rate constant (min$^{-1}$); and $\tau$ is the time required for 50% adsorbate breakthrough (min).

*2.7. Adsorbed Column Regeneration Experiments*

The regeneration experiments were performed with the AZT-saturated column at the highest η conditions: packed-bed height of 13 cm, [AZT]$_{\text{inlet}}$ = 2 mg L$^{-1}$, Q = 1 mL min$^{-1}$ with pH 5.0. The experiments were performed in order to verify the adsorbent regeneration capacity. The column regeneration was performed using AZT-free water (deionized water), at the same flow rate fed in upward mode, immediately after the saturation point was established, in a total of 5 cycles of adsorption–desorption, according to [30].

## 3. Results and Discussion

### 3.1. Adsorbent Characterization

The microscopy image of the adsorbent obtained from SEM is shown in Figure 2. The analysis of Figure 2 revealed a porous structure characteristic of the *Moringa oleifera* Lam [31] seed, mainly composed of cellulose, fibers, and lignin [24], and the highly porous surface observed is a feature that enhances the adsorption process [32]. In addition, Figure 2 shows that the porous structure is asymmetric and heterogeneous, which is often observed for adsorbents that undergo chemical and/or thermal treatments [33].

The analysis of the FTIR spectra (Figure 3) revealed the functional groups identified on the surface of the adsorbent before and after the adsorption process. The low-intensity peak observed at 1097 cm$^{-1}$ was attributed to the O–H and/or C–O stretching [34] and at 1056 cm$^{-1}$ to the C–O stretching of alcohol and phenol groups, both of which were observed only before the adsorption process took place. At 1234 cm$^{-1}$, the peak was attributed to the COOH group present in the lignin molecule [35]. The ranges 1453–1513 cm$^{-1}$ and 1654–1742 cm$^{-1}$ were assigned to the C=C stretching vibration and the C=O carbonyl stretching vibration, respectively [36], even though [33] associated the stretching vibrations of C=C with a peak near 1640 cm$^{-1}$. The peak intensity observed before the adsorption was attributed to a negatively charged carboxylic group (–COO$^-$) bond stretching, and after the adsorption, it was related to the N–H deformation and C=N bending vibrations contained in the ATZ heterocyclic structure [23,37]. The peaks intensity at 1854 cm$^{-1}$ and 2922 cm$^{-1}$ were attributed to the C–H stretching vibration of aliphatic chains CH$_3$ and CH$_2$, respectively [38]. Finally, the peak near 3394 cm$^{-1}$ was due to the hydrogen vibrations of O–H bonds [39].

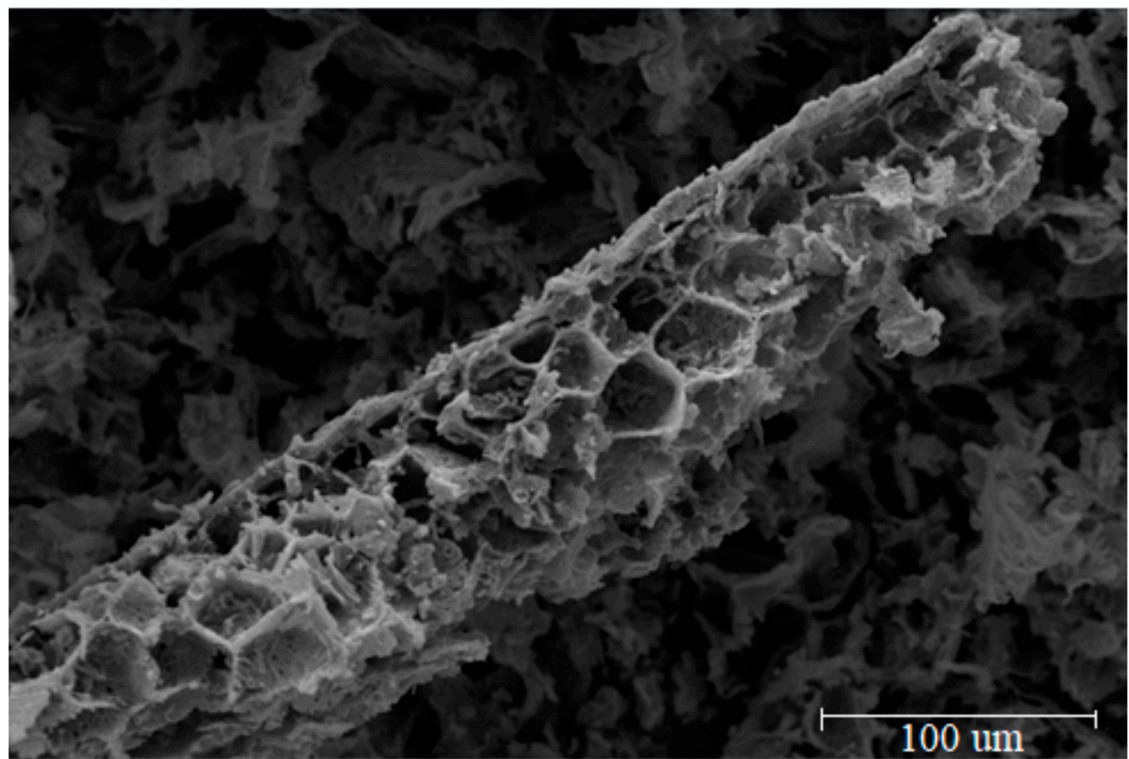

**Figure 2.** Microscopy image obtained from SEM for the adsorbent prepared from *Moringa oleifera* Lam. seeds.

Figure 3 shows the FTIR spectra of the adsorbent before and after the column adsorption of ATZ using *Moringa oleifera* Lam. seeds as adsorbents.

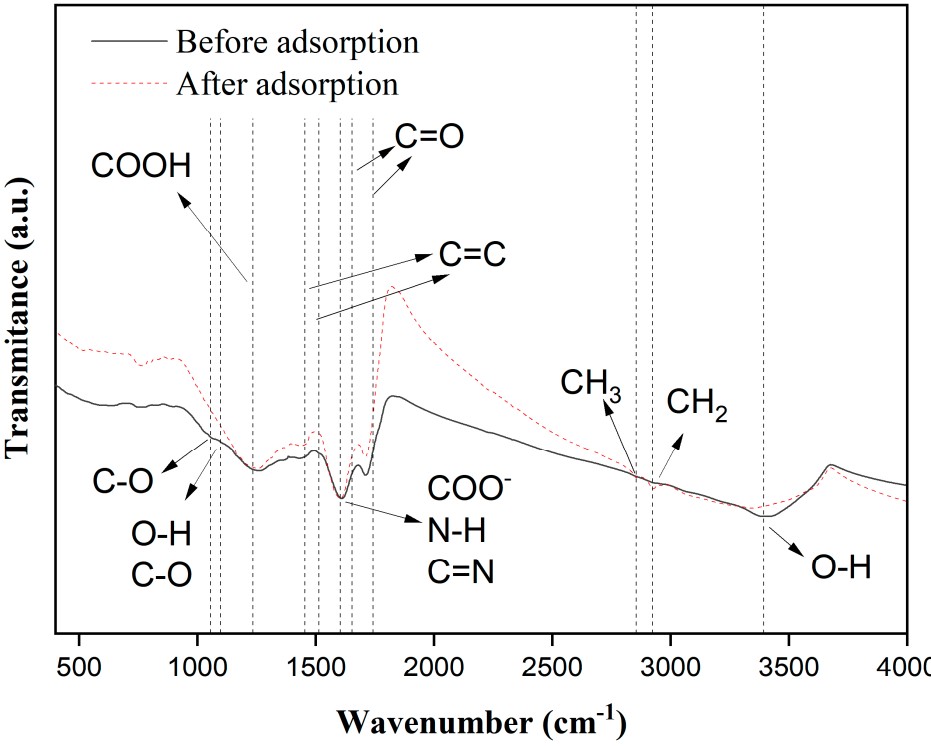

**Figure 3.** FTIR spectra of the adsorbent before and after the column adsorption of ATZ using *Moringa oleifera* Lam. seeds as adsorbents.

The changes in peak intensities (Figure 3) before and after the adsorption process suggested the presence of electrostatic interaction (i) and hydrogen bonding (ii) as the two mechanisms of ATZ adsorption at the molecular level. Mechanism (i) is due to the positively charged amine groups in the ATZ molecule with the negatively charged carboxylic groups at the adsorbent surface; mechanism (ii) is related to the H-bond between the amine in the ATZ contaminant and oxygenated groups in the adsorbent surface such as carboxylic acid, alcohol, and others. Both mechanisms were suggested by other authors as well, such as: [33] working with ATZ adsorption using *Moringa oleifera* pods, [23] using the seed husks from *Moringa oleifera* Lam for ATZ adsorption, and the review of *Moringa oleifera* seeds in water treatment provided by [40], which presented different mechanisms of adsorption on *Moringa oleifera* for different contaminants.

The diffractogram of the *Moringa oleifera* Lam. seeds and of the synthesized adsorbent is shown in Figure 4.

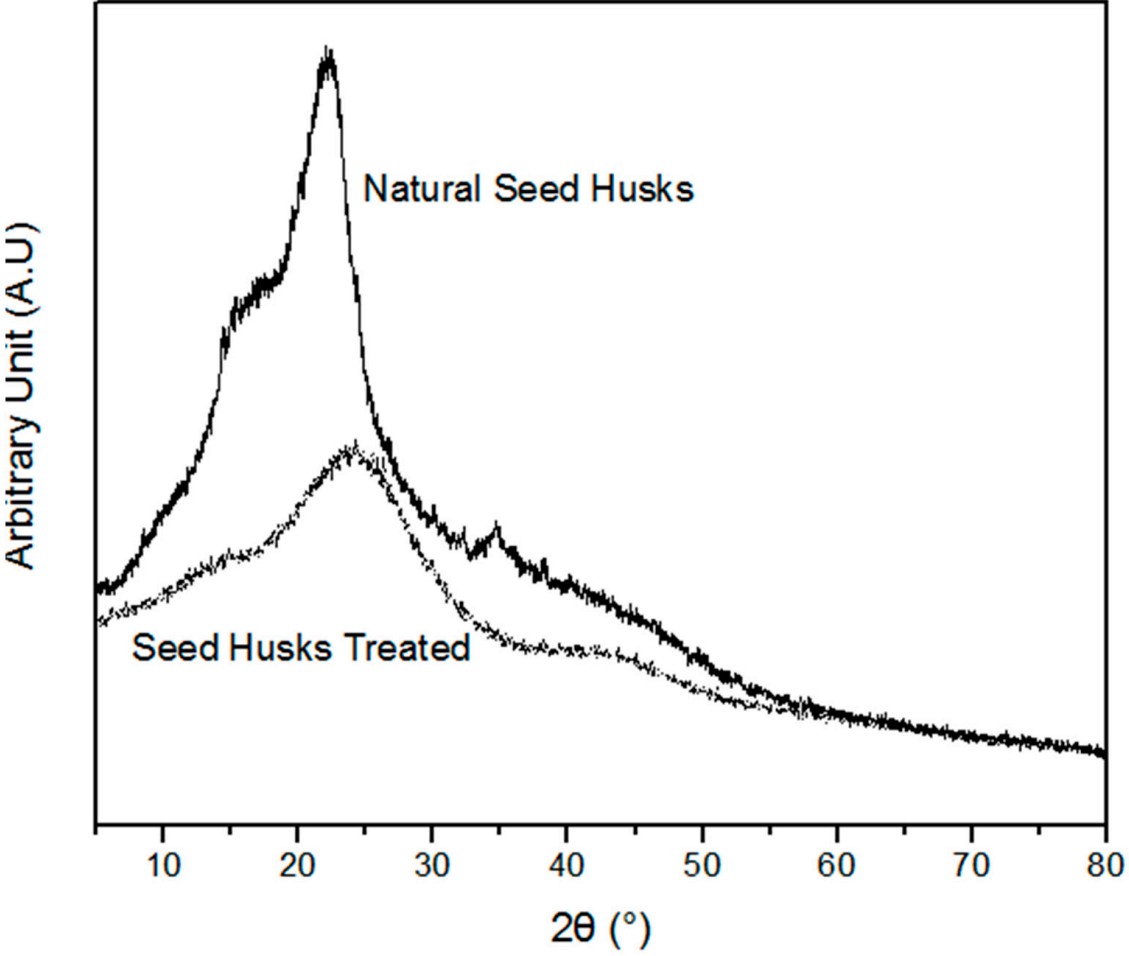

**Figure 4.** XRD diffractogram of the *Moringa oleifera* Lam. seeds and of the adsorbent synthesized.

The results presented in Figure 4 revealed that the natural *Moringa oleifera* Lam, seeds presented two characteristic diffraction interactions, one at approximately $2\theta = 22°$ and the other at approximately $2\theta = 34°$, which are characteristics of cellulose type I materials [41]. The adsorbent synthesized from *Moringa oleifera* Lam. seeds presented a broad diffraction characteristic of amorphous materials in the region of approximately $18° < 2\theta < 28°$ [42].

The texture properties of the low-cost adsorbent synthesized from *Moringa oleifera* Lam. seeds are displayed in Table 1.

The data presented in Table 1 revealed that the synthesized adsorbent increased the BET-specific surface area by 37% in comparison to the *Moringa oleifera* Lam. seeds, but the

average pore diameter decreased from ~36 Å to ~29 Å. An increase in the BET-specific surface is desired since it favors the adsorption process [43]. The surface area increase was due to the chemical/thermal process that provided an ~70% increase in the total pore volume (Table 1).

**Table 1.** Texture properties of *Moringa oleifera* Lam. seeds and of the synthesized low-cost adsorbent.

|  | *Moringa oleifera* **Lam. Seeds** | **Adsorbent** |
| --- | --- | --- |
| BET-specific surface area ($m^2\ g^{-1}$) | 1.822 | 2.496 |
| Average pore diameter (Å) | 36.28 | 29.46 |
| Total pore volume ($cm^3\ g^{-1}$) | 0.0256 | 0.0437 |
| Micropore volume ($cm^3\ g^{-1}$) | 0.0041 | 0.0274 |
| Mesopore volume ($cm^3\ g^{-1}$) | 0.0215 | 0.0163 |

In Figure 5, the results obtained from the zeta potential analysis for the synthesized low-cost adsorbent are shown. The pH varied from 2–12, with the pH of the isoelectric point being found at $pH_{IEP}$ = 4.8, meaning that the adsorbent possesses a net negative charge at a pH higher than pH > 4.8 and a net positive charge for pH < 4.8 [44,45].

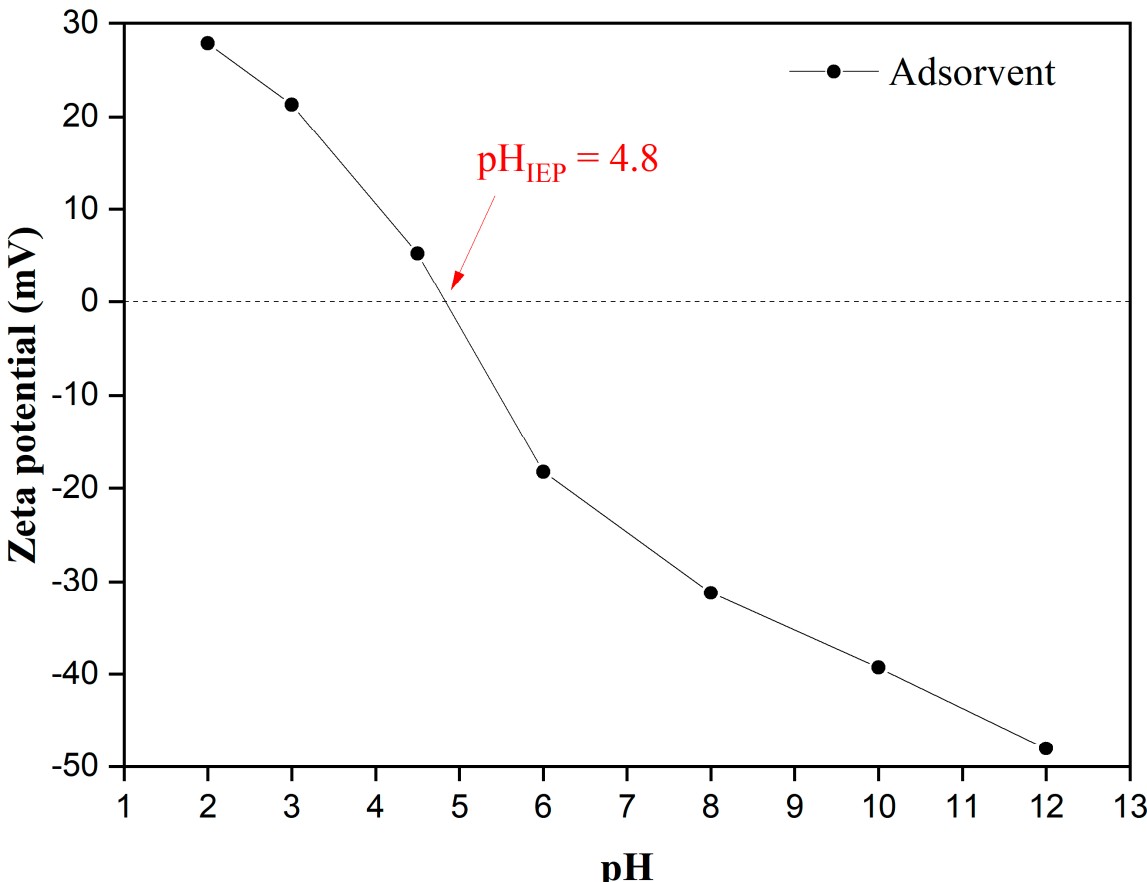

**Figure 5.** Zeta potential of the low-cost adsorbent synthesized from *Moringa oleifera* Lam. seeds.

### 3.2. Column Adsorption—Experimental Data

The experimental breakthrough curves of the ATZ adsorption in a packed-bed column using *Moringa oleifera* Lam. seeds as adsorbent were displayed in Figure 6, in which the effects of the peristaltic pump flow rate (Figure 6a), bed height (Figure 6b), and [ATZ]$_{inlet}$ (Figure 6c) can be seen.

The data and parameters from the breakthrough curves for the ATZ adsorption in a packed-bed column using *Moringa oleifera* Lam. seeds as adsorbent are summarized in Table 2.

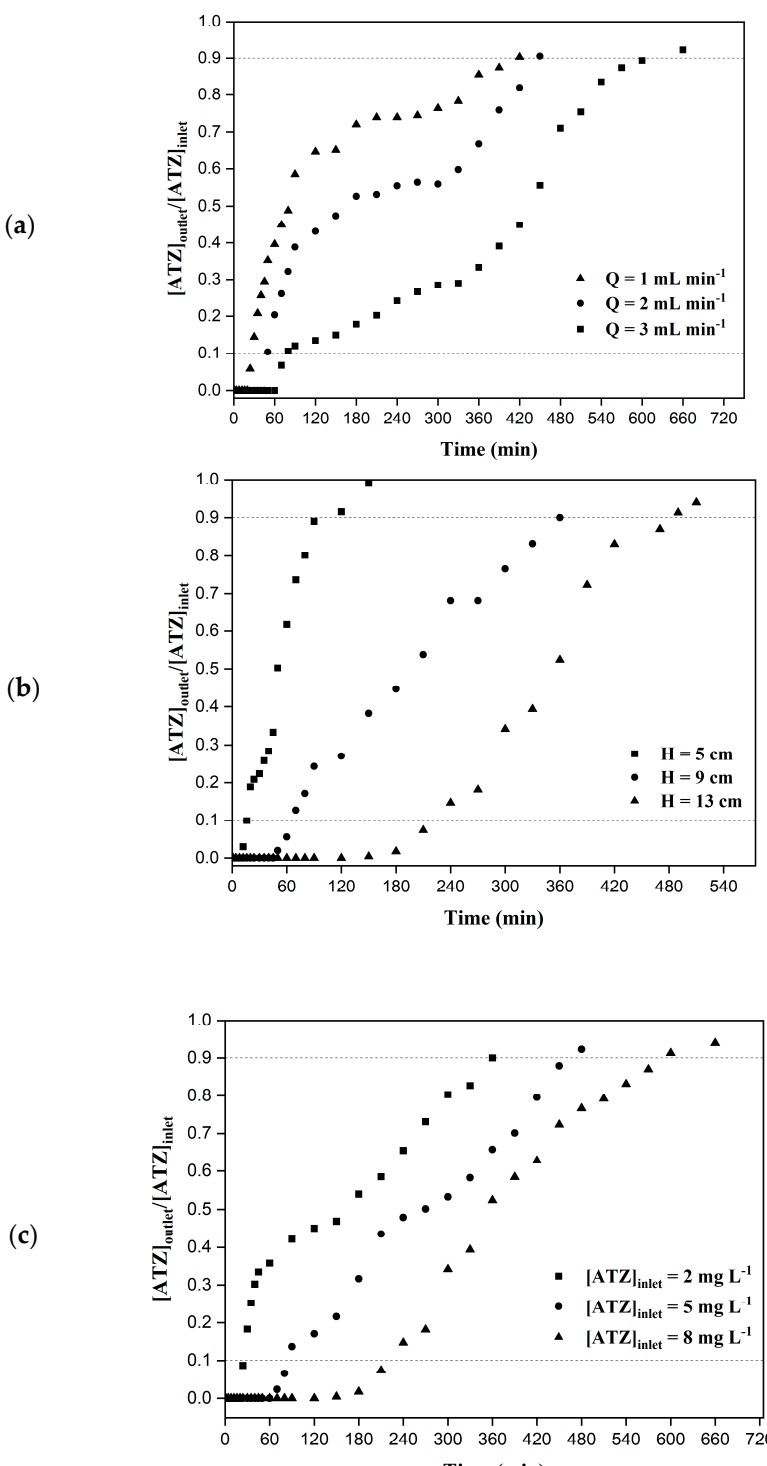

**Figure 6.** Breakthrough curves for the ATZ adsorption in a packed-bed column using *Moringa oleifera* Lam. seeds as adsorbent: (**a**) effect of peristaltic pump flow rate with solution pH 5.0, fixed bed height (13 cm), and $[ATZ]_{inlet} = 2$ mg L$^{-1}$ at T = 25 °C; (**b**) effect of the bed height with $[ATZ]_{inlet} = 2$ mg L$^{-1}$, flow rate of 1 mL min$^{-1}$, and pH 5.0; (**c**) effect of the inlet ATZ concentration with flow rate fixed at 1 mL min$^{-1}$, pH 5.0, and bed height of 13 cm.

**Table 2.** Parameters of the breakthrough curves for the ATZ adsorption in a packed-bed column using *Moringa oleifera* Lam. seeds as adsorbent.

| H (cm) | Q (mL/min) | pH | [ATZ]$_{inlet}$ (mg/L) | $t_b$ (min) | $t_s$ (min) | $q_e$ (mg/g) | $q_b$ (mg/g) | $W_t$ (mg) | η (%) |
|---|---|---|---|---|---|---|---|---|---|
| 13 | 1.0 | 5.0 | 2.0 | 25 | 420 | 1.51 | 1.460 | 2.92 | 50,00 |
| 13 | 2.0 | 5.0 | 2.0 | 49 | 453 | 0.98 | 0.760 | 1.68 | 45.23 |
| 13 | 3.0 | 5.0 | 2.0 | 78 | 601 | 0.73 | 0.326 | 0.84 | 38.80 |
| 5 | 1.0 | 5.0 | 2.0 | 14 | 97 | 0.26 | 0.08 | 0.26 | 30.70 |
| 9 | 1.0 | 5.0 | 2.0 | 65 | 360 | 0.54 | 0.128 | 0.37 | 34.59 |
| 13 | 1.0 | 5.0 | 2.0 | 218 | 485 | 1.62 | 1.41 | 2.96 | 47.79 |
| 13 | 1.0 | 5.0 | 2.0 | 27 | 360 | 1.38 | 1.27 | 2.63 | 48.28 |
| 13 | 1.0 | 5.0 | 5.0 | 85 | 474 | 1.05 | 0.69 | 2.04 | 33.82 |
| 13 | 1.0 | 5.0 | 8.0 | 221 | 601 | 0.87 | 0.32 | 1.55 | 20.64 |

Notes: $t_b$: breakthrough time (min); $t_s$: saturation time (min); $q_e$: maximum adsorption capacity (mg g$^{-1}$); $W_t$: total mass of Atrazine transported through the adsorption column (mg); η: percentage of Atrazine removed.

From Figure 6a, it was observed that an increase in the ATZ inlet flow rate (Q) was followed by an increase in the slope of the breakthrough curves, i.e., an increase from decreased $t_b$ and $t_s$, as displayed in Table 2. In addition, an increase from Q = 1 mL/min up to 3 mL/min decreased the percentage of ATZ removed (η) from 50% to ~39% (Table 2), which was expected and corroborated by other authors [46]. From Figure 6b, an increase in the bed height (H), from 5 cm to 13 cm, shifted the breakthrough curves in the right direction due to an increase in the breakthrough time ($t_b$, 14 min to 218 min) and saturation time ($t_s$, 97 min to 485 min). This behavior can be explained by the increase in bed height and adsorbent mass inside the column, which, in turn, increased the adsorption capacity since there are a greater number of available sites for the adsorption of ATZ. This trend was also observed in the ATZ removal in a packed-bed column reactor with biogenic manganese oxide (BMO) in which the ATZ removal increased from 20 to 39% when the BMO content increased 6-fold [47]. From Figure 6c, an increase in the [ATZ]$_{inlet}$ was followed by a rapid saturation of the packed-bed column due to an increase in the amount of ATZ molecules going through the column and adsorbing onto the material available site, in agreement with the bifenthrin pesticide biosorption in a fixed-bed column that reduced the saturation time of the column when a concentrated pesticide solution was flowing through the column [48].

The highest ATZ removal observed was η = 50% (Table 2), which can be associated with the sharpness (or S-shape) of the breakthrough curve [49] displayed in Figure 6. The sharper the breakthrough, the higher the expected removal since it reflects low mass transport resistance [50]. The curvature shown in the experimental data (Figure 6) suggests that different mass transport mechanisms are important for ATZ column adsorption such as micro-, macropore, and surface resistances [51]. The determination of each mass transfer parameter can be tedious and time-consuming; therefore, these parameters are commonly lumped into different models such as the logistic model and used for an empirical analysis of the mass transfer mechanisms involved.

*3.3. Column Adsorption—Model Fitting*

The logistic model fitting for the packed-bed column adsorption of ATZ using *Moringa oleifera* Lam. seeds as adsorbent is shown in Figure 7, in which the effects of the peristaltic pump flow rate (Figure 7a), bed height (Figure 7b), and [ATZ]$_{inlet}$ (Figure 7c) can be observed.

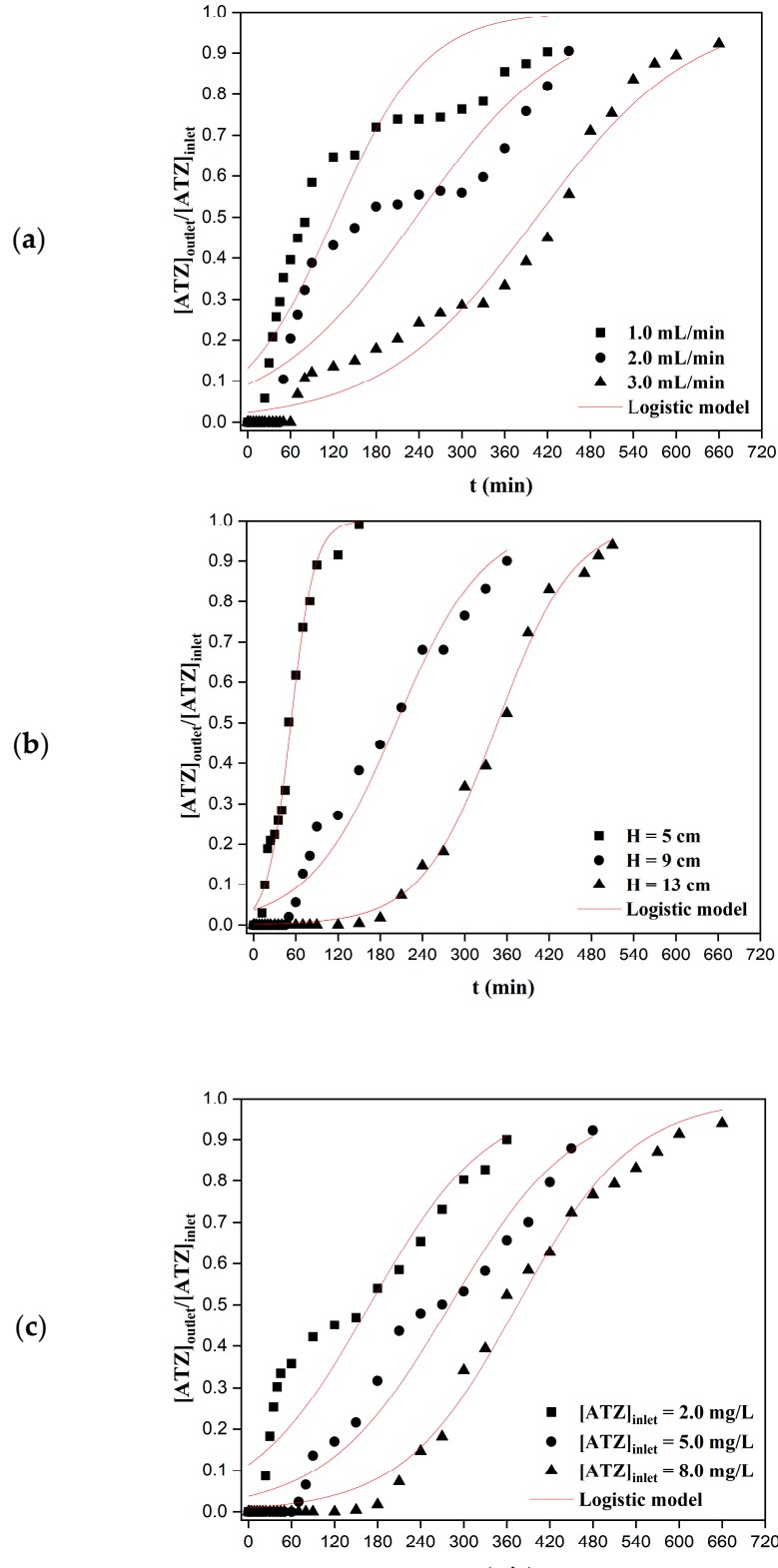

**Figure 7.** Logistic model fitting for the packed-bed column adsorption of ATZ using *Moringa oleifera* Lam. seeds as adsorbent for: (**a**) different peristaltic pump flow rate with solution pH 5.0, fixed bed height (13 cm), and $[ATZ]_{inlet} = 2$ mg $L^{-1}$ at T = 25 °C; (**b**) different bed heights with $[ATZ]_{inlet} = 2$ mg $L^{-1}$, flow rate of 1 mL $min^{-1}$, and pH 5.0; (**c**) effect of the inlet ATZ concentration with flow rate fixed at 1 mL $min^{-1}$, pH 5.0, and bed height of 13 cm.

The parameters of the models fitting used in Figure 7 and the statistical data are compiled in Table 3.

**Table 3.** Parameters of the models fitted to the experimental data operating the packed-bed adsorption column.

| Model | Parameter | Q (mL min$^{-1}$) | | | | H (cm) | | [ATZ]$_{inlet}$ (mg L$^{-1}$) | | |
|---|---|---|---|---|---|---|---|---|---|---|
| | | 1 | 2 | 3 | 5 | 9 | 13 | 2 | 5 | 8 |
| Logistic | a | 1.8857 | 2.2934 | 3.7343 | 3.1690 | 3.2597 | 6.3883 | 2.0452 | 3.2493 | 4.667 |
| | b | 0.0156 | 0.0097 | 0.0092 | 0.0596 | 0.0161 | 0.0184 | 0.0121 | 0.0116 | 0.0124 |
| | $R^2$ | 0.8399 | 0.8563 | 0.9758 | 0.9843 | 0.9668 | 0.9963 | 0.886 | 0.9664 | 0.9907 |
| | RMSE | 0.0175 | 0.0132 | 0.0023 | 0.0019 | 0.0032 | 0.0004 | 0.0107 | 0.0033 | 0.0011 |
| Bohart–Adams | $k_{BA}$ (L mg$^{-1}$ min$^{-1}$) | $7.54 \times 10^{-3}$ | $4.69 \times 10^{-3}$ | $4.44 \times 10^{-3}$ | 0.0288 | $7.78 \times 10^{-3}$ | $8.89 \times 10^{-3}$ | $6.05 \times 10^{-3}$ | $2.32 \times 10^{-3}$ | $1.55 \times 10^{-3}$ |
| | $N_0$ (mg L$^{-1}$) | 52.66 | 205.99 | 530.47 | 23.163 | 49.00 | 58.17 | 27.36 | 113.36 | 243.71 |
| Thomas | $k_T$ (mL mg$^{-1}$ min$^{-1}$) | 0.0075 | 0.0047 | 0.0044 | 0.0288 | 0.0078 | 0.0089 | 0.0061 | 0.0023 | 0.0016 |
| | $q_e$ (mg g$^{-1}$) | 208.5 | 815.6 | 2100.5 | 91.7 | 231.5 | 292.1 | 137.4 | 569.3 | 1223.9 |
| Yoon–Nelson | $k_{YN}$ (min$^{-1}$) | 0.0156 | 0.0097 | 0.0092 | 0.0596 | 0.0161 | 0.0184 | 0.0121 | 0.0116 | 0.0124 |
| | $\tau$ (min) | 120.8 | 236.4 | 405.9 | 53.1 | 202.4 | 347.1 | 169.0 | 280.1 | 376.3 |

Notes: a: intercept of the logistic model; b: slope of the logistic model; $R^2$: coefficient of determination; RMSE: root-mean-square error; $k_{BA}$: Bohart–Adams kinetic constant (L mg$^{-1}$ min$^{-1}$); $N_0$: saturation concentration (mg L$^{-1}$); H: bed height (cm); $k_T$: Thomas model rate constant (L mg$^{-1}$ min$^{-1}$); $k_{YN}$: Yoon–Nelson rate constant (min$^{-1}$); $\tau$: time required for 50% adsorbate breakthrough (min).

The technology of continuous adsorption has been investigated in the last few years and has provided useful insights and advances in the removal of contaminants of emerging concern from water and wastewater [52]. However, the Thomas model, Yoon–Nelson model, and Bohart–Adams model are all mathematically equivalent, i.e., even though they have differences in their considerations, hypotheses, and physical meaning, from an exclusive mathematical point of view, they are all derived from the logistic model, thus it is not possible to determine different statistical parameters ($R^2$, for example) for each of them [29]. Therefore, the best model fitted to the experimental data of ATZ adsorption in a packed-bed column was the logistic model as shown in Table 3, with the best fit ($R^2 = 0.9963$, RMSE = 0.0004) observed for the column packed with a 13 cm bed height, [ATZ]$_{inlet}$ = 2 mg L$^{-1}$, flow rate of 1 mL min$^{-1}$, and pH of 5.0.

Interpreting the continuous adsorption by means of the Bohart–Adams model, the breakthrough curves can be explained for the initial part, i.e., [ATZ]$_{outlet}$/[ATZ]$_{inlet}$ < 1. In the initial part of the breakthrough curve, the values of the $N_0$ parameter increased for all variables (Q, H, and [ATZ]$_{inlet}$), and the parameter $k_{BA}$ decreased for all variables (Q, H, and [ATZ]$_{inlet}$), meaning that the adsorption initial rate is limited by the external mass transfer [53]. The Thomas model is one of the most used for modeling continuous adsorption experiments. The increase in flow rate increased the parameter $q_e$ and decreased $k_T$, meaning a rapid saturation [54]. This was also observed for increasing bed height and ATZ inlet concentration, meaning insufficient residence time [55]. An outlook with the Yoon–Nelson model, a simple and theoretical continuous adsorption model, shows that the values of $k_{YN}$ decreased for all variables (Q, H, and [ATZ]$_{inlet}$) but $\tau$ increased, which was not expected. Since η represents the time required for 50% adsorbate breakthrough, it is supposed to decrease with increasing flow rate and [ATZ]$_{inlet}$ and to decrease with greater bed height [56]. Therefore, the Yoon–Nelson model is not a good fit for our results.

## 4. Conclusions

In this study, the objective was to prepare a low-cost adsorbent from *Moringa oleifera* Lam. seeds in order to perform the continuous adsorption of the herbicide Atrazine from a synthetically contaminated water sample using a packed-bed column. The adsorbent synthesis revealed an increase of ~37% in the specific surface area and a ~70% increase in the total pore volume. The effect of the ATZ inlet flow rate decreased the percentage

of ATZ removed (η) due to a lower residence time for the ATZ molecules, but the bed height increased the η value due to the availability of a greater number of adsorption sites. However, the [ATZ] inlet also decreased the value of η due to external mass transfer resistance at the initial adsorption time. The highest percentage removal of ATZ (η = 50, 0%) was observed with a packed-bed column with H = 13 cm bed height, Q = 1 mL/min, $[ATZ]_{inlet}$ = 2.0 mg/L, pH = 5.0, and a breakthrough time of 25 min and saturation time of 420 min. The Bohart–Adams, Thomas, and Yoon–Nelson models were used to analyze the modeling of the experimental data, but all the models' parameters were derived from the logistic model due to the mathematical identity of these equations. In conclusion, this paper showed the effect of bed height, flow rate, and contaminant inlet solution on the adsorption of the herbicide Atrazine in a continuous operation using a packed-bed column with a low-cost adsorbent prepared from *Moringa oleifera* Lam. Seeds, showing effective removal values for this particular emerging contaminant of difficult removal.

**Author Contributions:** R.B. conceptualization, formal analysis, investigation, data curation, writing—original draft, project administration; D.M. methodology, data curation; A.D. writing—original draft, project administration; C.d.O.B. resources, writing—review and editing, supervision; H.B.Q. conceptualization, formal analysis, investigation, data curation, writing—original draft, project administration; G.W. conceptualization, formal analysis; M.R.F.-K. resources, writing—review and editing; L.F.C. resources, writing—review and editing, supervision. All authors have read and agreed to the published version of the manuscript.

**Funding:** This research received no external funding.

**Acknowledgments:** The authors acknowledge the financial support of the National Council for Scientific and Technological Development (CNPq) and the Coordination of Superior Level Staff Improvement (Capes).

**Conflicts of Interest:** The authors declare no conflict of interest.

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
