# Peer review of "Adsorption of Atrazine from Synthetic Contaminated Water Using a Packed-Bed Column with a Low-Cost Adsorbent (Moringa oleifera Lam.)"

_water, doi:10.3390/w15071260_

Round 1
Reviewer 1 Report
Comments for authors
1/ The necessity and the novelty of this study should be clearly in the manuscript ?
2/ What are the potential health and environmental risks associated with the use of ATZ?
3/ What are some of the techniques documented in the literature that have been used to remove ATZ from contaminated water?
4/ Adsorbent characterization is an important step in determining its effectiveness for removing contaminants from water or other solutions. Transmission Electron Microscopy (TEM) is a technique used to study the morphology and size of the adsorbent particles. X-ray photoelectron spectroscopy (XPS) is used to identify the chemical composition of the adsorbent's surface. BET (Brunauer-Emmett-Teller) is a technique used to measure the specific surface area of the adsorbent, which is important for determining the adsorption capacity of the material. By using a combination of these techniques, researchers can obtain a comprehensive understanding of the adsorbent's properties and its potential for use in water treatment applications.
5/ Could the authors have provided more information on the mechanism of atrazine adsorption by Moringa oleifera Lam in their study? Specifically, what factors contribute to the adsorption process and how does Moringa oleifera Lam interact with atrazine at a molecular level?
6/ Could the authors incorporate recent literature from the past five years, such as the article "Photocatalyst and sustainability of water" published in Arabian Journal for Science and Engineering (2021) and the study on desalination and water treatment published in the Journal of Desalination and Water Treatment (2020)? These articles provide valuable insights into the use of photocatalysts and sustainable water treatment methods and can be cited to enhance the relevance and accuracy of the manuscript.
Author Response
Thank you.

Reviewer 2 Report
Dear Authors, please see attached PDF file.

Author Response
Obrigado.

Round 2
Reviewer 1 Report
No comments
Author Response
There are no comments for authors from the noble reviewer #1. We would like to thank noble referee for reading our reviewed manuscript and for all suggestions provided. It certainly helped to improve the quality of the paper.
Reviewer 2 Report
Dear Authors, thank you for the update of your manuscript.
Equations 7, 8, and 9 are slightly shifted to the left. Please, do correction.
Thank you!
Author Response
Thank you noble referee 2 for all suggestions made in the first and second round of revision. Eqs 1, 7, 8 and 9 alignment were adjusted.
